# Algorithmic Stability and Uniform Generalization

**Ibrahim Alabdulmohsin**
King Abdullah University of Science and Technology
Thuwal 23955, Saudi Arabia
`ibrahim.alabdulmohsin@kaust.edu.sa`

## Abstract

One of the central questions in statistical learning theory is to determine the conditions under which agents can learn from experience. This includes the necessary and sufficient conditions for generalization from a given finite training set to new observations. In this paper, we prove that algorithmic stability in the inference process is *equivalent* to uniform generalization across all parametric loss functions. We provide various interpretations of this result. For instance, a relationship is proved between stability and data processing, which reveals that algorithmic stability can be improved by post-processing the inferred hypothesis or by augmenting training examples with artificial noise prior to learning. In addition, we establish a relationship between algorithmic stability and the size of the observation space, which provides a formal justification for dimensionality reduction methods. Finally, we connect algorithmic stability to the size of the hypothesis space, which recovers the classical PAC result that the size (complexity) of the hypothesis space should be controlled in order to improve algorithmic stability and improve generalization.

## 1 Introduction

One fundamental goal of any learning algorithm is to strike a right balance between *underfitting* and *overfitting*. In mathematical terms, this is often translated into two separate objectives. First, we would like the learning algorithm to produce a hypothesis that is reasonably consistent with the empirical evidence (i.e. to have a small empirical risk). Second, we would like to guarantee that the empirical risk (training error) is a valid estimate of the true unknown risk (test error). The former condition protects against underfitting while the latter condition protects against overfitting.

The rationale behind these two objectives can be understood if we define the *generalization risk* $R_{gen}$ by the absolute difference between the empirical and true risks: $R_{gen} \doteq |R_{emp} - R_{true}|$. Then, it is elementary to observe that the true risk $R_{true}$ is bounded from above by the sum $R_{emp} + R_{gen}$. Hence, by minimizing both the empirical risk (underfitting) and the generalization risk (overfitting), one obtains an inference procedure whose true risk is minimal.

Minimizing the empirical risk alone can be carried out using the *empirical risk minimization* (ERM) procedure [1] or some approximations to it. However, the generalization risk is often impossible to deal with directly. Instead, it is a common practice to bound it *analyticaly* so that we can establish conditions under which it is guaranteed to be small. By establishing conditions for generalization, one hopes to design better learning algorithms that both perform well empirically and generalize well to novel observations in the future. A prominent example of such an approach is the Support Vector Machines (SVM) algorithm for binary classification [2].

However, bounding the generalization risk is quite intricate because it can be approached from various angles. In fact, several methods have been proposed in the past to prove generalization bounds including uniform convergence, algorithmic stability, Rademacher and Gaussian complexities, generic chaining bounds, the PAC-Bayesian framework, and robustness-based analysis

[1, 3, 4, 5, 6, 7, 8, 9]. Concentration of measure inequalities form the building blocks of these rich theories.

The proliferation of generalization bounds can be understood if we look into the general setting of learning introduced by Vapnik [1]. In this setting, we have an observation space $\mathcal{Z}$ and a hypothesis space $\mathcal{H}$. A learning algorithm, henceforth denoted $\mathcal{L} : \cup_{m=1}^{\infty} \mathcal{Z}^m \rightarrow \mathcal{H}$, uses a finite set of observations to infer a hypothesis $H \in \mathcal{H}$. In the general setting, the inference process end-to-end is influenced by three key factors: (1) the nature of the observation space $\mathcal{Z}$, (2) the nature of the hypothesis space $\mathcal{H}$, and (3) the details of the learning algorithm $\mathcal{L}$. By imposing constraints on any of these three components, one may be able to derive new generalization bounds. For example, the Vapnik-Chervonenkis (VC) theory derives generalization bounds by assuming constraints on $\mathcal{H}$, while stability bounds, e.g. [6, 10, 11, 12], are derived by assuming constraints on $\mathcal{L}$.

Given that different generalization bounds can be established by imposing constraints on any of $\mathcal{Z}$, $\mathcal{H}$, or $\mathcal{L}$, it is intriguing to ask if there exists a single view for generalization that ties all of these different components together. In this paper, we answer this question in the affirmative by establishing that algorithmic stability alone is equivalent to *uniform* generalization. Informally speaking, an inference process is said to generalize uniformly if the generalization risk vanishes uniformly across *all* bounded parametric loss functions at the limit of large training sets. A more precise definition will be presented in the sequel. We will show why constraints that are imposed on either $\mathcal{H}$, $\mathcal{Z}$, or $\mathcal{L}$ to improve uniform generalization can be interpreted as methods of improving the stability of the learning algorithm $\mathcal{L}$. This is similar in spirit to a result by Kearns and Ron, who showed that having a finite VC dimension in the hypothesis space $\mathcal{H}$ implies a certain notion of algorithmic stability in the inference process [13]. Our statement, however, is more general as it applies to all learning algorithms that fall under Vapnik's general setting of learning, well beyond uniform convergence.

The rest of the paper is as follows. First, we review the current literature on algorithmic stability, generalization, and learnability. Then, we introduce key definitions that will be repeatedly used throughout the paper. Next, we prove the central theorem, which reveals that algorithmic stability is equivalent to uniform generalization, and provide various interpretations of this result afterward.

## 2 Related Work

Perhaps, the two most fundamental concepts in statistical learning theory are those of *learnability* and *generalization* [12, 14]. The two concepts are distinct from each other. As will be discussed in more details next, whereas learnability is concerned with measuring the *excess risk* within a hypothesis space, generalization is concerned with estimating the *true risk*.

In order to define learnability and generalization, suppose we have an observation space $\mathcal{Z}$, a probability distribution of observations $\mathbb{P}(z)$, and a bounded stochastic loss function $L(\cdot; H) : \mathcal{Z} \rightarrow [0, 1]$, where $H \in \mathcal{H}$ is an inferred hypothesis. Note that $L$ is implicitly a function of (parameterized by) $H$ as well. We define the true risk of a hypothesis $H \in \mathcal{H}$ by the risk functional:

$$R_{true}(H) = \mathbb{E}_{Z \sim \mathbb{P}(z)} \big[ L(Z; H) \big] \tag{1}$$

Then, a learning algorithm is called *consistent* if the true risk of its inferred hypothesis $H$ converges to the optimal true risk within the hypothesis space $\mathcal{H}$ at the limit of large training sets $m \rightarrow \infty$. A problem is called *learnable* if it admits a consistent learning algorithm [14]. It has been known that learnability for supervised classification and regression problems is equivalent to uniform convergence [3, 14]. However, Shalev-Shwartz et al. recently showed that uniform convergence is not necessary in Vapnik's general setting of learning and proposed algorithmic stability as an alternative key condition for learnability [14].

Unlike learnability, the question of generalization is concerned primarily with how representative the empirical risk $R_{emp}$ is to the true risk $R_{true}$. To elaborate, suppose we have a finite training set $S_m = \{Z_i\}_{i=1,..,m}$, which comprises of $m$ i.i.d. observations $Z_i \sim \mathbb{P}(z)$. We define the empirical risk of a hypothesis $H$ with respect to $S_m$ by:

$$R_{emp}(H; S_m) = \frac{1}{m} \sum_{Z_i \in S_m} L(Z_i; H) \tag{2}$$

We also let $R_{true}(H)$ be the true risk as defined in Eq. (1). Then, a learning algorithm $\mathcal{L}$ is said to generalize if the empirical risk of its inferred hypothesis converges to its true risk as $m \rightarrow \infty$.

Similar to learnability, uniform convergence is, by definition, sufficient for generalization [1], but it is not necessary because the learning algorithm can always restrict its search space to a smaller subset of $\mathcal{H}$ (artificially so to speak). By contrast, it is not known whether algorithmic stability is necessary for generalization. It has been shown that various notions of algorithmic stability can be defined that are sufficient for generalization [6, 10, 11, 12, 15, 16]. However, it is not known whether an appropriate notion of algorithmic stability can be defined that is *both* necessary and sufficient for generalization in Vapnik's general setting of learning. In this paper, we answer this question by showing that stability in the inference process is not only sufficient for generalization, but it is, in fact, *equivalent to uniform generalization*, which is a notion of generalization that is stronger than the one traditionally considered in the literature.

## 3 Preliminaries

To simplify the discussion, we will always assume that all sets are countable, including the observation space $\mathcal{Z}$ and the hypothesis space $\mathcal{H}$. This is similar to the assumptions used in some previous works such as [6]. However, the main results, which are presented in Section 4, can be readily generalized. In addition, we assume that all learning algorithms are invariant to permutations of the training set. Hence, the order of training examples is irrelevant.

Moreover, if $X \sim \mathbb{P}(x)$ is a random variable drawn from the alphabet $\mathcal{X}$ and $f(X)$ is a function of $X$, we write $\mathbb{E}_{X \sim \mathbb{P}(x)} f(X)$ to mean $\sum_{x \in \mathcal{X}} \mathbb{P}(x) f(x)$. Often, we will simply write $\mathbb{E}_X f(X)$ to mean $\mathbb{E}_{X \sim \mathbb{P}(x)} f(X)$ if the distribution of $X$ is clear from the context. If $X$ takes its values from a finite set $S$ uniformly at random, we write $X \sim S$ to denote this distribution of $X$. If $X$ is a boolean random variable, then $\mathbb{I}\{X\} = 1$ if and only if $X$ is true, otherwise $\mathbb{I}\{X\} = 0$. In general, random variables are denoted with capital letters, instances of random variables are denoted with small letters, and alphabets are denoted with calligraphic typeface. Also, given two probability mass functions $P$ and $Q$ defined on the same alphabet $\mathcal{A}$, we will write $\langle P, \ Q \rangle$ to denote the *overlapping coefficient*, i.e. intersection, between $P$ and $Q$. That is, $\langle P, \ Q \rangle \doteq \sum_{a \in \mathcal{A}} \min\{P(a), Q(a)\}$. Note that $\langle P, \ Q \rangle = 1 - ||P, \ Q||_{\mathcal{T}}$, where $||P, \ Q||_{\mathcal{T}}$ is the total variation distance. Last, we will write $B(k; \phi, n) = \binom{n}{k} \phi^k (1 - \phi)^{n-k}$ to denote the binomial distribution.

In this paper, we consider the general setting of learning introduced by Vapnik [1]. To reiterate, we have an observation space $\mathcal{Z}$ and a hypothesis space $\mathcal{H}$. Our learning algorithm $\mathcal{L}$ receives a set of $m$ observations $S_m = \{Z_i\}_{i=1,...,m} \in \mathcal{Z}^m$ generated i.i.d. from a fixed unknown distribution $\mathbb{P}(z)$, and picks a hypothesis $H \in \mathcal{H}$ with probability $\mathbb{P}_{\mathcal{L}}(H = h | S_m)$. Formally, $\mathcal{L} : \cup_{m=1}^{\infty} \mathcal{Z}^m \to \mathcal{H}$ is a stochastic map. In this paper, we allow the hypothesis $H$ to be any *summary statistic* of the training set. It can be a measure of central tendency, as in unsupervised learning, or it can be a mapping from an input space to an output space, as in supervised learning. In fact, we even allow $H$ to be a subset of the training set itself. In formal terms, $\mathcal{L}$ is a stochastic map between the two random variables $H \in \mathcal{H}$ and $S_m \in \mathcal{Z}^m$, where the exact interpretation of those random variables is irrelevant.

In any learning task, we assume a non-negative bounded loss function $L(Z; H) : \mathcal{Z} \to [0, 1]$ is used to measure the quality of the inferred hypothesis $H \in \mathcal{H}$ on the observation $Z \in \mathcal{Z}$. Most importantly, we assume that $L(\cdot; H) : \mathcal{Z} \to [0, 1]$ is *parametric*:

**Definition 1** (Parametric Loss Functions). *A loss function $L(\cdot; H) : \mathcal{Z} \to [0, 1]$ is called parametric if it is independent of the training set $S_m$ given the inferred hypothesis $H$. That is, a parametric loss function satisfies the Markov chain: $S_m \to H \to L(\cdot; H)$.*

For any fixed hypothesis $H \in \mathcal{H}$, we define its true risk $R_{true}(H)$ by Eq. (1), and define its empirical risk on a training set $S_m$, denoted $R_{emp}(H; S_m)$, by Eq. (2). We also define the true and empirical risks of the *learning algorithm* $\mathcal{L}$ by the expected risk of its inferred hypothesis:

$$\hat{R}_{true}(\mathcal{L}) = \mathbb{E}_{S_m} \mathbb{E}_{H \sim \mathbb{P}_{\mathcal{L}}(h | S_m)} R_{true}(H) \qquad = \mathbb{E}_{S_m} \mathbb{E}_{H | S_m} R_{true}(H) \qquad (3)$$

$$\hat{R}_{emp}(\mathcal{L}) = \mathbb{E}_{S_m} \mathbb{E}_{H \sim \mathbb{P}_{\mathcal{L}}(h | S_m)} R_{emp}(H; S_m) \qquad = \mathbb{E}_{S_m} \mathbb{E}_{H | S_m} R_{emp}(H; S_m) \qquad (4)$$

To simplify notation, we will write $\hat{R}_{true}$ and $\hat{R}_{emp}$ instead of $\hat{R}_{true}(\mathcal{L})$ and $\hat{R}_{emp}(\mathcal{L})$. We will consider the following definition of generalization:

**Definition 2** (Generalization). *A learning algorithm $\mathcal{L} : \cup_{m=1}^{\infty} \mathcal{Z}^m \to \mathcal{H}$ with a parametric loss function $L(\cdot; H) : \mathcal{Z} \to [0, 1]$ generalizes if for any distribution $\mathbb{P}(z)$ on $\mathcal{Z}$, we have $\lim_{m \to \infty} |\hat{R}_{emp} - \hat{R}_{true}| = 0$, where $\hat{R}_{true}$ and $\hat{R}_{emp}$ are given in Eq. (3) and Eq. (4) respectively.*

In other words, a learning algorithm $\mathcal{L}$ generalizes according to Definition 2 if its empirical performance (training loss) becomes an unbiased estimator to the true risk as $m \to \infty$. Next, we define uniform generalization:

**Definition 3** (Uniform Generalization). *A learning algorithm $\mathcal{L} : \cup_{m=1}^{\infty} \mathcal{Z}^m \to \mathcal{H}$ generalizes uniformly if for any $\epsilon > 0$, there exists $m_0(\epsilon) > 0$ such that for all distributions $\mathbb{P}(z)$ on $\mathcal{Z}$, all parametric loss functions, and all sample sizes $m > m_0(\epsilon)$, we have $\left| \hat{R}_{emp}(\mathcal{L}) - \hat{R}_{true}(\mathcal{L}) \right| \leq \epsilon$.*

Uniform generalization is stronger than the original notion of generalization in Definition 2. In particular, if a learning algorithm generalizes uniformly, then it generalizes according to Definition 2 as well. The converse, however, is not true. Even though uniform generalization appears to be quite a strong condition, at first sight, a key contribution of this paper is to show that it is *not* a strong condition because it is equivalent to a simple condition, namely *algorithmic stability*.

## 4 Main Results

Before we prove that algorithmic stability is equivalent to uniform generalization, we introduce a probabilistic notion of *mutual stability* between two random variables. In order to abstract away any *labeling* information the random variables might possess, e.g. the observation space may or may not be a metric space, we define stability by the impact of observations on *probability distributions*:

**Definition 4** (Mutual Stability). *Let $X \in \mathcal{X}$ and $Y \in \mathcal{Y}$ be two random variables. Then, the mutual stability between $X$ and $Y$ is defined by:*

$$S(X; Y) \doteq \langle \mathbb{P}(X) \, \mathbb{P}(Y), \ \mathbb{P}(X, Y) \rangle = \mathbb{E}_X \langle \mathbb{P}(Y), \ \mathbb{P}(Y|X) \rangle = \mathbb{E}_Y \langle \mathbb{P}(X), \ \mathbb{P}(X|Y) \rangle$$

If we recall that $0 \leq \langle P, \ Q \rangle \leq 1$ is the overlapping coefficient between the two probability distributions $P$ and $Q$, we see that $S(X; Y)$ given by Definition 4 is indeed a probabilistic measure of mutual stability. It measures how stable the distribution of $Y$ is before and after observing an instance of $X$, and vice versa. A small value of $S(X; Y)$ means that the probability distribution of $X$ or $Y$ is *heavily perturbed by a single observation* of the other random variable. Perfect mutual stability is achieved when the two random variables are independent of each other.

With this probabilistic notion of mutual stability in mind, we define the stability of a learning algorithm $\mathcal{L}$ by the mutual stability between its inferred hypothesis and a random training example.

**Definition 5** (Algorithmic Stability). *Let $\mathcal{L} : \cup_{m=1}^{\infty} \mathcal{Z}^m \to \mathcal{H}$ be a learning algorithm that receives a finite set of training examples $S_m = \{Z_i\}_{i=1,...,m} \in \mathcal{Z}^m$ drawn i.i.d. from a fixed distribution $\mathbb{P}(z)$. Let $H \sim \mathbb{P}_{\mathcal{L}}(h|S_m)$ be the hypothesis inferred by $\mathcal{L}$, and let $Z_{trn} \sim S_m$ be a single random training example. We define the stability of $\mathcal{L}$ by: $\mathbb{S}(\mathcal{L}) = \inf_{\mathbb{P}(z)} S(H; Z_{trn})$, where the infimum is taken over all possible distributions of observations $\mathbb{P}(z)$. A learning algorithm is called algorithmically stable if $\lim_{m \to \infty} \mathbb{S}(\mathcal{L}) = 1$.*

Note that the above definition of algorithmic stability is rather weak; it only requires that the contribution of any *single* training example on the overall inference process to be more and more negligible as the sample size increases. In addition, it is well-defined even if the learning algorithm is deterministic because the hypothesis $H$, if it is a deterministic function of an entire training set of $m$ observations, remains a *stochastic* function of any *individual* observation. We illustrate this concept with the following example:

**Example 1.** *Suppose that observations $Z_i \in \{0, 1\}$ are i.i.d. Bernoulli trials with $\mathbb{P}(Z_i = 1) = \phi$, and that the hypothesis produced by $\mathcal{L}$ is the empirical average $H = \frac{1}{m} \sum_{i=1}^{m} Z_i$. Because $\mathbb{P}(H = k/m \,|\, Z_{trn} = 1) = B(k-1; \phi, m-1)$ and $\mathbb{P}(H = k/m \,|\, Z_{trn} = 0) = B(k; \phi, m-1)$, it can be shown using Stirling's approximation [17] that the algorithmic stability of this learning algorithm is asymptotically given by $\mathbb{S}(\mathcal{L}) \sim 1 - \frac{1}{\sqrt{2 \pi m}}$, which is achieved when $\phi = 1/2$. A more general statement will be proved later in Section 5.*

Next, we show that the notion of algorithmic stability in Definition 5 is equivalent to the notion of uniform generalization in Definition 3. Before we do that, we first state the following lemma.

**Lemma 1** (Data Processing Inequality). *Let $A, B,$ and $C$ be three random variables that satisfy the Markov chain $A \to B \to C$. Then: $S(A; B) \leq S(A; C)$.*

*Proof.* The proof consists of two steps [1]. First, we note that because the Markov chain implies that $\mathbb{P}(C|B, A) = \mathbb{P}(C|B)$, we have $S(A; (B,C)) = S(A; B)$ by direct substitution into Definition 5. Second, similar to the *information-cannot-hurt* inequality in information theory [18], it can be shown that $S(A; (B,C)) \leq S(A; C)$ for any random variables $A$, $B$ and $C$. This is proved using some algebraic manipulation and the fact that the minimum of the sums is always larger than the sum of minimums, i.e. $\min\left\{\sum_i \alpha_i, \sum_i \beta_i\right\} \geq \sum_i \min\{\alpha_i, \beta_i\}$. Combining both results yields $S(A; B) = S(A; (B,C)) \leq S(A; C)$, which is the desired result. $\square$

Now, we are ready to state the main result of this paper.

**Theorem 1.** *For any learning algorithm $\mathcal{L} : \cup_{m=1}^{\infty} \mathcal{Z}^m \to \mathcal{H}$, algorithmic stability as given in Definition 5 is both necessary and sufficient for uniform generalization (see Definition 3). In addition, $\left|\hat{R}_{true} - \hat{R}_{emp}\right| \leq 1 - S(H; Z_{trn}) \leq 1 - \mathbb{S}(\mathcal{L})$, where $R_{true}$ and $R_{emp}$ are the true and empirical risks of the learning algorithm defined in Eq. (3) and (4) respectively.*

*Proof.* Here is an outline of the proof. First, because a parametric loss function $L(\cdot; H) : \mathcal{Z} \to [0, 1]$ is itself a random variable that satisfies the Markov chain $S_m \to H \to L(\cdot; H)$, it is not independent of $Z_{trn} \sim S_m$. Hence, the empirical risk is given by $\hat{R}_{emp} = \mathbb{E}_{L(\cdot;H)} \mathbb{E}_{Z_{trn}|L(\cdot;H)} L(Z_{trn}; H)$. By contrast, the true risk is given by $\hat{R}_{true} = \mathbb{E}_{L(\cdot;H)} \mathbb{E}_{Z_{trn} \sim \mathbb{P}(z)} L(Z_{trn}; H)$. The difference is:

$$\hat{R}_{true} - \hat{R}_{emp} = \mathbb{E}_{L(\cdot;H)} \left[\mathbb{E}_{Z_{trn}} L(Z_{trn}; H) - \mathbb{E}_{Z_{trn}|L(\cdot;H)} L(Z_{trn}; H)\right]$$

To sandwich the right-hand side between an upper and a lower bound, we note that if $\mathbb{P}_1(z)$ and $\mathbb{P}_2(z)$ are two distributions defined on the same alphabet $\mathcal{Z}$ and $F(\cdot) : \mathcal{Z} \to [0, 1]$ is a bounded loss function, then $\left|\mathbb{E}_{Z \sim \mathbb{P}_1(z)} F(Z) - \mathbb{E}_{Z \sim \mathbb{P}_2(z)} F(Z)\right| \leq ||\mathbb{P}_1(z), \mathbb{P}_2(z)||_{\mathcal{T}}$, where $||P, Q||_{\mathcal{T}}$ is the total variation distance. The proof to this result can be immediately deduced by considering the two regions $\{z \in \mathcal{Z} : \mathbb{P}_1(z) > \mathbb{P}_2(z)\}$ and $\{z \in \mathcal{Z} : \mathbb{P}_1(z) < \mathbb{P}_2(z)\}$ separately. This is, then, used to deduce the inequalities:

$$\left|\hat{R}_{true} - \hat{R}_{emp}\right| \leq 1 - S(L(\cdot; H); Z_{trn}) \leq 1 - S(H; Z_{trn}) \leq 1 - \mathbb{S}(\mathcal{L}),$$

where the second inequality follows by the data processing inequality in Lemma 1, whereas the last inequality follows by definition of algorithmic stability (see Definition 5). This proves that if $\mathcal{L}$ is algorithmically stable, i.e. $\mathbb{S}(\mathcal{L}) \to 1$ as $m \to \infty$, then $\left|\hat{R}_{true} - \hat{R}_{emp}\right|$ converges to zero uniformly across all parametric loss functions. Therefore, algorithmic stability is sufficient for uniform generalization. The converse is proved by showing that for any $\delta > 0$, there exists a bounded parametric loss and a distribution $\mathbb{P}_\delta(z)$ such that $1 - \mathbb{S}(\mathcal{L}) - \delta \leq \left|\hat{R}_{true} - \hat{R}_{emp}\right| \leq 1 - \mathbb{S}(\mathcal{L})$. Therefore, algorithmic stability is also necessary for uniform generalization. $\square$

## 5 Interpreting Algorithmic Stability and Uniform Generalization

In this section, we provide several interpretations of algorithmic stability and uniform generalization. In addition, we show how Theorem 1 recovers some classical results in learning theory.

### 5.1 Algorithmic Stability and Data Processing

The relationship between algorithmic stability and data processing is presented in Lemma 1. Given the random variables $A$, $B$, and $C$ and the Markov chain $A \to B \to C$, we always have $S(A; B) \leq S(A; C)$. This presents us with qualitative insights into the design of machine learning algorithms.

First, suppose we have two different hypotheses $H_1$ and $H_2$. We will say that $H_2$ contains *less informative* than $H_1$ if the Markov chain $S_m \to H_1 \to H_2$ holds. For example, if observations $Z_i \in \{0, 1\}$ are Bernoulli trials, then $H_1 \in \mathbb{R}$ can be the empirical average as given in Example 1 while $H_2 \in \{0, 1\}$ can be the label that occurs most often in the training set. Because $H_2 = \mathbb{I}\{H_1 \geq m/2\}$, the hypothesis $H_2$ contains strictly less information about the original training set than $H_1$. Formally, we have $S_m \to H_1 \to H_2$. In this case, $H_2$ enjoys a better *uniform* generalization bound than $H_1$ because of data-processing. Intuitively, we know that such a result should hold because $H_2$ is less tied to the original training set than $H_1$. This brings us to the following remark.

**Remark 1.** *We can improve the uniform generalization bound (or equivalently algorithmic stability) of a learning algorithm by post-processing its inferred hypothesis $H$ in a manner that is conditionally independent of the original training set given $H$.*

**Example 2.** *Post-processing hypotheses is a common technique used in machine learning. This includes sparsifying the coefficient vector $w \in \mathbb{R}^d$ in linear methods, where $w_j$ is set to zero if it has a small absolute magnitude. It also includes methods that have been proposed to reduce the number of support vectors in SVM by exploiting linear dependence [19]. By the data processing inequality, such methods improve algorithmic stability and uniform generalization.*

Needless to mention, better generalization does not immediately translate into a smaller true risk. This is because the empirical risk itself may increase when the inferred hypothesis is post-processed *independently* of the original training set.

Second, if the Markov chain $A \to B \to C$ holds, we also obtain $S(A; C) \geq S(B; C)$ by applying the data processing inequality to the *reverse* Markov chain $C \to B \to A$. As a result, we can improve algorithmic stability by contaminating training examples with artificial noise *prior* to learning. This is because if $\hat{S}_m$ is a perturbed version of a training set $S_m$, then $S_m \to \hat{S}_m \to H$ implies that $S(Z_{trn}; H) \geq S(\hat{Z}_{trn}; H)$, when $Z_{trn} \sim S_m$ and $\hat{Z}_{trn} \sim \hat{S}_m$ are random training examples drawn uniformly at random from each training set respectively. This brings us to the following remark:

**Remark 2.** *We can improve the algorithmic stability of a learning algorithm by introducing artificial noise to training examples, and applying the learning algorithm on the perturbed training set.*

**Example 3.** *Corrupting training examples with artificial noise, such as the recent dropout method, are popular techniques in neural networks to improve generalization [20]. By the data processing inequality, such methods indeed improve algorithmic stability and uniform generalization.*

## 5.2 Algorithmic Stability and the Size of the Observation Space

Next, we look into how the size of the observation space $\mathcal{Z}$ influences algorithmic stability. First, we start with the following definition:

**Definition 6** (Lazy Learning). *A learning algorithm $\mathcal{L}$ is called* lazy *if its hypothesis $H \in \mathcal{H}$ is mapped one-to-one with the training set $S_m$, i.e. the mapping $H \to S_m$ is injective.*

A lazy learner is called lazy if its hypothesis is *equivalent* to the original training set in its information content. Hence, no learning actually takes place. One example is instance-based learning when $H = S_m$. Despite their simple nature, lazy learners are useful in practice. They are useful theoretical tools as well. In particular, because of the equivalence $H \equiv S_m$ and the data processing inequality, the algorithmic stability of a lazy learner provides a *lower bound* to the stability of *any* possible learning algorithm. Therefore, we can relate algorithmic stability (uniform generalization) to the size of the observation space by quantifying the algorithmic stability of lazy learners. Because the size of $\mathcal{Z}$ is usually infinite, however, we introduce the following definition of *effective* set size.

**Definition 7.** *In a countable space $\mathcal{Z}$ endowed with a probability mass function $\mathbb{P}(z)$, the effective size of $\mathcal{Z}$ w.r.t. $\mathbb{P}(z)$ is defined by:* $\mathbf{Ess}\left[\mathcal{Z}; \mathbb{P}(z)\right] \doteq 1 + \left(\sum_{z \in \mathcal{Z}} \sqrt{\mathbb{P}(z)\,(1 - \mathbb{P}(z))}\right)^2$.

At one extreme, if $\mathbb{P}(z)$ is *uniform* over a finite alphabet $\mathcal{Z}$, then $\mathbf{Ess}\left[\mathcal{Z}; \mathbb{P}(z)\right] = |\mathcal{Z}|$. At the other extreme, if $\mathbb{P}(z)$ is a Kronecker delta distribution, then $\mathbf{Ess}\left[\mathcal{Z}; \mathbb{P}(z)\right] = 1$. As proved next, this notion of effective set size *determines* the rate of convergence of an empirical probability mass function to its true distribution when the distance is measured in the total variation sense. As a result, it allows us to relate algorithmic stability to a property of the observation space $\mathcal{Z}$.

**Theorem 2.** *Let $\mathcal{Z}$ be a countable space endowed with a probability mass function $\mathbb{P}(z)$. Let $S_m$ be a set of $m$ i.i.d. samples $Z_i \sim \mathbb{P}(z)$. Define $\mathbb{P}_{S_m}(z)$ to be the empirical probability mass function induced by drawing samples uniformly at random from $S_m$. Then: $\mathbb{E}_{S_m} ||\mathbb{P}(z), \mathbb{P}_{S_m}(z)||_\mathcal{T} = \sqrt{\frac{\mathbf{Ess}\left[\mathcal{Z}; \mathbb{P}(z)\right] - 1}{2\,\pi\,m}} + o(1/\sqrt{m})$, where $1 \leq \mathbf{Ess}\left[\mathcal{Z}; \mathbb{P}(z)\right] \leq |\mathcal{Z}|$ is the effective size of $\mathcal{Z}$ (see Definition 7). In addition, for any learning algorithm $\mathcal{L}: \cup_{m=1}^{\infty} \mathcal{Z}^m \to \mathcal{H}$, we have $S(H; Z_{trn}) \geq 1 - \sqrt{\frac{\mathbf{Ess}\left[\mathcal{Z}; \mathbb{P}(z)\right] - 1}{2\,\pi\,m}} - o(1/\sqrt{m})$, where the bound is achieved by lazy learners (see Definition 6)[2].*

*Proof.* Here is an outline of the proof. First, we know that $\mathbb{P}(S_m) = \binom{m}{m_1, m_2, \ldots} p_1^{m_1} p_2^{m_2} \cdots$, where $\binom{\cdot}{\cdot}$ is the multinomial coefficient. Using the relation $||P, Q||_{\mathcal{T}} = \frac{1}{2}||P - Q||_1$, the multinomial series, and *De Moivre's formula* for the mean deviation of the binomial random variable [22], it can be shown with some algebraic manipulations that:

$$\mathbb{E}_{S_m} ||\mathbb{P}(z), \mathbb{P}_{S_m}(z)||_{\mathcal{T}} = \frac{1}{m} \sum_{k=1,2,\ldots} (1-p_k)^{(1-p_k)m} p_k^{1+mp_k} \frac{m!}{(p_k m)! \left((1-p_k)m - 1\right)!}$$

Using *Stirling's approximation* to the factorial [17], we obtain the simple asymptotic expression:

$$\mathbb{E}_{S_m} ||\mathbb{P}(z), \mathbb{P}_{S_m}(z)||_{\mathcal{T}} \sim \frac{1}{2} \sum_{k=1,2,3,\ldots} \sqrt{\frac{2p_k(1-p_k)}{\pi m}} = 1 - \sqrt{\frac{\mathbf{Ess}\left[\mathcal{Z}; \mathbb{P}(z)\right] - 1}{2\pi m}},$$

which is tight due to the tightness of the Stirling approximation. The rest of the theorem follows from the Markov chain $S_m \to S_m \to H$, the data processing inequality, and Definition 6. $\qquad\square$

**Corollary 1.** *Given the conditions of Theorem 2, if $\mathcal{Z}$ is in addition finite (i.e. $|\mathcal{Z}| < \infty$), then for any learning algorithm $\mathcal{L}$, we have:* $\mathbb{S}(\mathcal{L}) \geq 1 - \sqrt{\frac{|\mathcal{Z}|-1}{2\pi m}} - o(1/\sqrt{m})$

*Proof.* Because in a finite observation space $\mathcal{Z}$, the maximum effective set size (see Definition 7) is $|\mathcal{Z}|$, which is attained at the uniform distribution $\mathbb{P}(z) = 1/|\mathcal{Z}|$. $\qquad\square$

Intuitively speaking, Theorem 2 and its corollary state that in order to guarantee good uniform generalization for *all* possible learning algorithms, the number of observations must be sufficiently large to cover the entire effective size of the observation space $\mathcal{Z}$. Needless to mention, this is difficult to achieve in practice so the algorithmic stability of machine learning algorithms must be controlled in order to guarantee a good generalization from a few empirical observations. Similarly, the uniform generalization bound can be improved by reducing the effective size of the observation space, such as by using dimensionality reduction methods.

## 5.3 Algorithmic Stability and the Complexity of the Hypothesis Space

Finally, we look into the hypothesis space and how it influences algorithmic stability. First, we look into the role of the *size* of the hypothesis space. This is formalized in the following theorem.

**Theorem 3.** *Denote by $H \in \mathcal{H}$ the hypothesis inferred by a learning algorithm $\mathcal{L} : \cup_{m=1}^{\infty} \mathcal{Z}^m \to \mathcal{H}$. Then, the following bound on algorithmic stability always holds:*

$$\mathbb{S}(\mathcal{L}) \geq 1 - \sqrt{\frac{\mathbf{H}(H)}{2\,m}} \geq 1 - \sqrt{\frac{\log |\mathcal{H}|}{2\,m}},$$

*where $\mathbf{H}$ is the Shannon entropy measured in nats (i.e. using natural logarithms).*

*Proof.* The proof is information-theoretic. If we let $I(X; Y)$ be the mutual information between the r.v.'s $X$ and $Y$ and let $S_m = \{Z_1, Z_2, \ldots, Z_m\}$ be a random choice of a training set, we have:

$$I(S_m; H) = \mathbf{H}(S_m) - \mathbf{H}(S_m \mid H) = \left[ \sum_{i=1}^{m} \mathbf{H}(Z_i) \right] - \left[ \mathbf{H}(Z_1|H) + \mathbf{H}(Z_2|Z_1, H) + \cdots \right]$$

Because conditioning reduces entropy, i.e. $\mathbf{H}(A|B) \leq \mathbf{H}(A)$ for any r.v.'s $A$ and $B$, we have:

$$I(S_m; H) \geq \sum_{i=1}^{m} \left[ \mathbf{H}(Z_i) - \mathbf{H}(Z_i \mid H) \right] = m \left[ \mathbf{H}(Z_{trn}) - \mathbf{H}(Z_{trn} \mid H) \right]$$

Therefore:

$$I(Z_{trn}; H) \leq \frac{I(S_m; H)}{m} \tag{5}$$

---

on average. This is believed to be the first appearance of the square-root law in statistical inference in the literature [21]. Because the effective set size of the Bernoulli distribution, according to Definition 7, is given by $1 + 4\phi(1 - \phi)$, Theorem 2 agrees with, in fact generalizes, de Moivre's result.

Next, we use *Pinsker's inequality* [18], which states that for any probability distributions $P$ and $Q$: $||P, \ Q||_\mathcal{T} \le \sqrt{\frac{D(P\,||\,Q)}{2}}$, where $||P, \ Q||_\mathcal{T}$ is total variation distance and $D(P\,||\,Q)$ is the Kullback-Leibler divergence measured in nats (i.e. using natural logarithms). If we recall that $S(S_m; H) = 1 - ||\mathbb{P}(S_m)\,\mathbb{P}(H), \ \mathbb{P}(S_m, H)||_\mathcal{T}$ while mutual information is $I(S_m; H) = D(\mathbb{P}(S_m, H)\,||\,\mathbb{P}(S_m)\,\mathbb{P}(H))$, we deduce from Pinsker's inequality and Eq. (5):

$$S(Z_{trn}; H) = 1 - ||\mathbb{P}(Z_{trn})\,\mathbb{P}(H), \ \mathbb{P}(Z_{trn}, H)||_\mathcal{T}$$

$$\ge 1 - \sqrt{\frac{I(Z_{trn}; H)}{2}} \ge 1 - \sqrt{\frac{I(S_m; H)}{2m}} \ge 1 - \sqrt{\frac{\mathbf{H}(H)}{2m}} \ge 1 - \sqrt{\frac{\log|\mathcal{H}|}{2m}}$$

In the last line, we used the fact that $I(X; Y) \le \mathbf{H}(X)$ for any random variables $X$ and $Y$. $\qquad\square$

Theorem 3 re-establishes the classical PAC result on the finite hypothesis space [23]. In terms of algorithmic stability, a learning algorithm will enjoy a high stability if the size of the hypothesis space is small. In terms of uniform generalization, it states that the generalization risk of a learning algorithm is bounded from above *uniformly* across all parametric loss functions by $\sqrt{\mathbf{H}(H)/(2m)} \le \sqrt{\log|\mathcal{H}|/(2m)}$, where $\mathbf{H}(H)$ is the Shannon entropy of $H$.

Next, we relate algorithmic stability to the Vapnik-Chervonenkis (VC) dimension. Despite the fact that the VC dimension is defined on binary-valued functions whereas algorithmic stability is a functional of probability distributions, there exists a connection between the two concepts. To show this, we first introduce a notion of an *induced concept class* that exists for any learning algorithm $\mathcal{L}$:

**Definition 8.** *The concept class $\mathcal{C}$ induced by a learning algorithm $\mathcal{L} : \cup_{m=1}^\infty \mathcal{Z}^m \to \mathcal{H}$ is defined to be the set of total Boolean functions $c(z) = \mathbb{I}\{\mathbb{P}(Z_{trn} = z \,|\, H) \ge \mathbb{P}(Z_{trn} = z)\}$ for all $H \in \mathcal{H}$.*

Intuitively, every hypothesis $H \in \mathcal{H}$ induces a total partition on the observation space $\mathcal{Z}$ given by the Boolean function in Definition 8. That is, $H$ splits $\mathcal{Z}$ into two disjoint sets: the set of values in $\mathcal{Z}$ that are, *a posteriori*, less likely to have been present in the training set than before given that the inferred hypothesis is $H$, and the set of all other values. The complexity (richness) of the induced concept class $\mathcal{C}$ is related to algorithmic stability via the VC dimension.

**Theorem 4.** *Let $\mathcal{L} : \cup_{m=1}^\infty \mathcal{Z}^m \to \mathcal{H}$ be a learning algorithm with an induced concept class $\mathcal{C}$. Let $d_{VC}(\mathcal{C})$ be the VC dimension of $\mathcal{C}$. Then, the following bound holds if $m > d_{VC}(\mathcal{C}) + 1$:*

$$\mathbb{S}(\mathcal{L}) \ge 1 - \frac{4 + \sqrt{d_{VC}(\mathcal{C})\,(1 + \log(2m))}}{\sqrt{2m}}$$

*In particular, $\mathcal{L}$ is algorithmically stable if its induced concept class $\mathcal{C}$ has a finite VC dimension.*

*Proof.* The proof relies on the fact that algorithmic stability $\mathbb{S}(\mathcal{L})$ is bounded from below by $1 - \sup_{\mathbb{P}(z)} \left\{ \mathbb{E}_{S_m} \ \sup_{h \in \mathcal{H}} \left| \mathbb{E}_{Z \sim \mathbb{P}(z)} \ c_h(Z) - \mathbb{E}_{Z \sim S_m} \ c_h(Z) \right| \right\}$, where $c_H(z) = \mathbb{I}\{\mathbb{P}(Z_{trn} = z|H) \ge \mathbb{P}(Z_{trn} = z)\}$. The final bound follows by applying uniform convergence results [23]. $\qquad\square$

## 6 Conclusions

In this paper, we showed that a probabilistic notion of algorithmic stability was equivalent to uniform generalization. In informal terms, a learning algorithm is called algorithmically stable if the impact of a single training example on the probability distribution of the final hypothesis always vanishes at the limit of large training sets. In other words, the inference process never depends heavily on any single training example. If algorithmic stability holds, then the learning algorithm generalizes well regardless of the choice of the parametric loss function. We also provided several interpretations of this result. For instance, the relationship between algorithmic stability and data processing reveals that algorithmic stability can be improved by either post-processing the inferred hypothesis or by augmenting training examples with artificial noise prior to learning. In addition, we established a relationship between algorithmic stability and the effective size of the observation space, which provided a formal justification for dimensionality reduction methods. Finally, we connected algorithmic stability to the complexity (richness) of the hypothesis space, which re-established the classical PAC result that the complexity of the hypothesis space should be controlled in order to improve stability, and, hence, improve generalization.

## Footnotes

[1]Detailed proofs are available in the supplementary file.

[2]A special case of Theorem 2 was proved by de Moivre in the 1730s, who showed that the empirical mean of i.i.d. Bernoulli trials with a probability of success $\phi$ converges to the true mean at a rate of $\sqrt{2\phi(1 - \phi)/(\pi m)}$

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
