[Supplementary Material · proofs.pdf]

# Algorithmic Stability and Uniform Generalization (Supplementary File)

**Ibrahim Alabdulmohsin**
King Abdullah University of Science and Technology
Thuwal 23955, Saudi Arabia
ibrahim.alabdulmohsin@kaust.edu.sa

*Note: The references cited in this document are referred to by their numbers as cited in the original paper.*

## 1 Proof of Example 1

There are two approaches to prove the asymptotic expression for algorithmic stability. The first approach is to look into the impact of knowing $H$ on the distribution of training examples $Z_{trn}$. The second (equivalent) approach is to look into the impact of a single training example $Z_{trn}$ on the final distribution of the inferred hypothesis $H$. Here, we use the first approach because it is simpler.

First, the probability we obtain a hypothesis $H = \frac{k}{m}$, where $k \in \{0, 1, \ldots, m\}$, given that we have $m$ Bernoulli trials has a binomial distribution:

$$\mathbb{P}(H = \frac{k}{m}) = \binom{m}{k} \phi^k (1-\phi)^{m-k}$$

We use the identity:

$$1 - S(H; Z_{trn}) = \big|\big|\mathbb{P}(H) \cdot \mathbb{P}(Z_{trn}), \ \mathbb{P}(H, Z_{trn})\big|\big|_{\mathcal{T}}$$
$$= \sum_{k=0}^{m} \mathbb{P}(H = \frac{k}{m}) \ \big|\big|\mathbb{P}(Z_{trn}), \ \mathbb{P}(Z_{trn} \,|\, H)\big|\big|_{\mathcal{T}},$$

where, again, $||P, Q||_{\mathcal{T}}$ is the total variation distance between the two probability distributions $P$ and $Q$.

However, $\mathbb{P}(Z_{trn})$ is Bernoulli with probability of success $\phi$ while $\mathbb{P}(Z_{trn} \,|\, H = \frac{k}{m})$ is Bernoulli with probability of success $H$. The total variation distance between the two Bernoulli distributions is given by $|\phi - H|$. So, we obtain:

$$1 - S(H; Z_{trn}) = \sum_{k=0}^{m} \binom{m}{k} \phi^k (1-\phi)^{m-k} \left|\phi - \frac{k}{m}\right| \tag{1}$$

This is the *mean deviation*. Assuming $\phi m$ is an integer, then the mean deviation of the binomial random variable is given by de Moivre's formula [22]:

$$MD = 2 (1-\phi)^{(1-\phi)\, m} \ \phi^{1+m\,\phi} (1 + m\,\phi) \binom{m}{m\,\phi + 1} \tag{2}$$

The mean deviation is maximized when $\phi = \frac{1}{2}$. This gives us:

$$\mathbb{S}(\mathcal{L}) = 1 - \frac{1}{2^m} \binom{m}{m/2 + 1} \sim 1 - \frac{1}{\sqrt{2\,\pi\,m}},$$

where in the last step we expanded the binomial coefficient and used Stirling's approximation [17].

## 2 Proof of Lemma 1

The proof consists of two steps. First, we show that $S(A; (B,C)) = S(A; B)$. To prove this, we note that:

$$S(A; (B,C))$$
$$= \sum_{A,B,C} \min\left\{\mathbb{P}(A)\,\mathbb{P}(B,C),\ \mathbb{P}(A,B,C)\right\}$$
$$= \sum_{A,B,C} \min\left\{\mathbb{P}(A)\,\mathbb{P}(B)\,\mathbb{P}(C|B),\ \mathbb{P}(A,B)\,\mathbb{P}(C|A,B)\right\}$$
$$= \sum_{A,B,C} \mathbb{P}(C|B)\,\min\left\{\mathbb{P}(A)\,\mathbb{P}(B),\ \mathbb{P}(A,B)\right\}$$
$$= \sum_{A,B} \min\left\{\mathbb{P}(A)\,\mathbb{P}(B),\ \mathbb{P}(A,B)\right\} = S(A;B)$$

In the third line, we used the Markov property $\mathbb{P}(C|B,A) = \mathbb{P}(C|B)$.

Second, we show that $S(A; (B,C)) \leq S(A; C)$ for any random variables $A, B$ and $C$. This is the analog to the *information-cannot-hurt* inequality in information theory. We have by definition:

$$S(A; (B,C)) = \sum_{A,B,C} \min\left\{\mathbb{P}(A)\,\mathbb{P}(B,C),\ \mathbb{P}(A,B,C)\right\}$$
$$= \sum_{A} \mathbb{P}(A) \sum_{B,C} \min\left\{\mathbb{P}(B,C),\ \mathbb{P}(B,C|A)\right\}$$

However, the minimum of the sums is always larger than the sum of minimums. That is:

$$\min\left\{\sum_i \alpha_i,\ \sum_i \beta_i\right\} \geq \sum_i \min\{\alpha_i,\ \beta_i\}$$

Using marginalization $\mathbb{P}(X) = \sum_Y \mathbb{P}(X, Y)$ and the above inequality, we obtain:

$$S(A; (B,C)) = \sum_{A} \mathbb{P}(A) \sum_{B,C} \min\left\{\mathbb{P}(B,C), \mathbb{P}(B,C|A)\right\}$$
$$\leq \sum_{A} \mathbb{P}(A) \sum_{C} \min\left\{\sum_{B} \mathbb{P}(B,C),\ \sum_{B} \mathbb{P}(B,C|A)\right\}$$
$$= \sum_{A,C} \min\{\mathbb{P}(A)\,\mathbb{P}(C), \mathbb{P}(A,C)\} = S(A;C)$$

Combining both results yields $S(A; B) = S(A; (B,C)) \leq S(A; C)$, which is the desired inequality.

## 3 Proof of Theorem 1

Let $\mathcal{L} : \cup_{m=1}^{\infty} \mathcal{Z}^m \to \mathcal{H}$ be a learning algorithm that receives a finite set of training examples $S_m = \{Z_i\}_{i=1,..,m} \in \mathcal{Z}^m$ drawn i.i.d. from a fixed unknown distribution $\mathbb{P}(z)$. Let $H \sim \mathbb{P}_{\mathcal{L}}(h|S_m)$ be a random variable that stands for the hypothesis inferred by $\mathcal{L}$, and let $Z_{trn} \sim S_m$ be a single random training example. To simplify notation, we will write $F = L(\cdot; H) : \mathcal{Z} \to [0,1]$ to denote the loss function whose dependence on $H$ is implicit. Note that $F$ is itself a random variable that satisfies the Markov chain $S_m \to H \to F$. The claim is that $\mathcal{L}$ generalizes uniformly across all parametric loss functions $F$ if and only if $\mathcal{L}$ is algorithmically stable.

By the Markov property, we have $\mathbb{P}(F|H, S_m) = \mathbb{P}(F|H)$. By definition, the true and empirical risks are given by:

$$\hat{R}_{true} = \mathbb{E}_{S_m} \mathbb{E}_{H|S_m} \mathbb{E}_{F|H} \mathbb{E}_{Z\sim\mathbb{P}(z)} F(Z)$$
$$= \mathbb{E}_F \mathbb{E}_{Z\sim\mathbb{P}(z)} F(Z) \tag{3}$$
$$\hat{R}_{emp} = \mathbb{E}_{S_m} \mathbb{E}_{H|S_m} \mathbb{E}_{F|H} \mathbb{E}_{Z\sim S_m} F(Z)$$
$$= \mathbb{E}_{S_m} \mathbb{E}_{F|S_m} \mathbb{E}_{Z\sim S_m} F(Z)$$
$$= \mathbb{E}_F \mathbb{E}_{S_m|F} \mathbb{E}_{Z\sim S_m} F(Z) \tag{4}$$

Because $Z_{trn} \sim S_m$ is a random variable whose value is chosen uniformly at random with replacement from the training set $S_m$, its original distribution *prior to* observing $F$ is the original distribution of observations $\mathbb{P}(z)$. Its *posterior* distribution after observation $F$ is altered, however, because both $F$ and $Z_{trn}$ depend on the choice of the training set $S_m$. However, they are both *conditionally* independent of each other given $S_m$. By marginalization, we have:

$$\mathbb{P}(Z_{trn}|F) = \mathbb{E}_{S_m|F} \mathbb{P}(Z_{trn}|S_m, F) = \mathbb{E}_{S_m|F} \mathbb{P}(Z_{trn}|S_m)$$

Combining this with Eq. (3) and Eq. (4) yields:

$$\hat{R}_{true} = \mathbb{E}_F \mathbb{E}_{Z_{trn}} F(Z_{trn})$$
$$\hat{R}_{emp} = \mathbb{E}_F \mathbb{E}_{Z_{trn}|F} F(Z_{trn}),$$

where in the first line $Z_{trn}$ is distributed according to its original distribution $\mathbb{P}(z)$. Both equations imply that:

$$\hat{R}_{true} - \hat{R}_{emp} = \mathbb{E}_F \left[ \mathbb{E}_{Z_{trn}} F(Z_{trn}) - \mathbb{E}_{Z_{trn}|F} F(Z_{trn}) \right]$$

Now, we would like to sandwich the right-hand side between upper and lower bounds. To do this, we note that if $\mathbb{P}_1(z)$ and $\mathbb{P}_2(z)$ are two distributions defined on the same alphabet $\mathcal{Z}$ and $F : \mathcal{Z} \to [0, 1]$ is a fixed bounded loss function, then:

$$\left| \mathbb{E}_{Z\sim\mathbb{P}_1(z)} F(Z) - \mathbb{E}_{Z\sim\mathbb{P}_2(z)} F(Z) \right| \leq ||\mathbb{P}_1(z), \ \mathbb{P}_2(z)||_{\mathcal{T}},$$

where $||P, \ Q||_{\mathcal{T}}$ is the total variation distance. The proof to this result can be immediately deduced by considering the two regions $\{z \in \mathcal{Z} : \mathbb{P}_1(z) > \mathbb{P}_2(z)\}$ and $\{z \in \mathcal{Z} : \mathbb{P}_1(z) < \mathbb{P}_2(z)\}$ separately. In addition, it is tight because the inequality holds with equality for the loss function $F(z) = \mathbb{I}\{\mathbb{P}_1(z) \geq \mathbb{P}_2(z)\}$.

Using the last result, we deduce the inequality:

$$\left| \hat{R}_{true} - \hat{R}_{emp} \right| \leq 1 - S(F; Z_{trn})$$

Finally, from the data processing inequality, we have $S(H; Z_{trn}) \leq S(F; Z_{trn})$. Plugging this into the earlier inequality yields the bound:

$$\left| \hat{R}_{true} - \hat{R}_{emp} \right| \leq 1 - S(H; Z_{trn})$$
$$\leq 1 - \mathbb{S}(\mathcal{L})$$

This proves that if $\mathcal{L}$ is algorithmically stable, i.e. $\mathbb{S}(\mathcal{L}) \to 1$ as $m \to \infty$, then $\left| \hat{R}_{true} - \hat{R}_{emp} \right|$ converge to zero uniformly across all parametric loss functions. Therefore, algorithmic stability is sufficient for uniform generalization.

To prove that algorithmic stability is also necessary for uniform generalization, let $\delta > 0$ be some fixed positive constant and let $\mathbb{P}_\delta(z)$ be a distribution of observations that achieves $S(H; Z_{trn}) < \mathbb{S}(\mathcal{L}) + \delta$, where $\mathbb{S}(\mathcal{L})$ is the algorithmic stability defined in the paper. Of course, such a probability distribution $\mathbb{P}_\delta(z)$ always exists by definition of the infimum in Definition 5. Next, let $L_\delta(\cdot; H) : \mathcal{Z} \to [0, 1]$ be a parametric loss that is given by:

$$L_\delta(z; H) = \mathbb{I}\{\mathbb{P}_\delta(Z_{trn} = z) \geq \mathbb{P}_\delta(Z_{trn} = z \,|\, H)\}$$
$$= \mathbb{I}\{\mathbb{P}_\delta(Z_{trn} = z) \geq \mathbb{E}_{S_m \,|\, H} \mathbb{P}_{Z_{trn}\sim S_m} (Z_{trn} = z)\}$$

The loss $L_\delta(\cdot; H)$ is independent of the training set given $H$ because $\mathbb{P}_\delta(Z_{trn} = z \,|\, H)$ is evaluated by taking expectation over all possible training sets given $H$. In addition, the loss function is parametric; it has a bounded range $L_\delta(\cdot; H) : \mathcal{Z} \to [0, 1]$ and satisfies the Markov chain $S_m \to H \to L_\delta(\cdot; H)$. However, given this choice of parametric loss, we have:

$$
\begin{aligned}
&|R_{true}(\mathcal{L}) - R_{emp}(\mathcal{L})| \\
&= \mathbb{E}_H \big[ \mathbb{E}_{Z_{trn}} \, \mathbb{I}\{\mathbb{P}_\delta(Z_{trn}) > \mathbb{P}_\delta(Z_{trn} \,|\, H)\} \; - \; \mathbb{E}_{Z_{trn} \,|\, H} \, \mathbb{I}\{\mathbb{P}_\delta(Z_{trn}) > \mathbb{P}_\delta(Z_{trn} \,|\, H)\} \big] \\
&= \mathbb{E}_H \sum_{Z_{trn}} \big[ \mathbb{P}(Z_{trn}) - \mathbb{P}(Z_{trn} \,|\, H) \big] \cdot \mathbb{I}\{\mathbb{P}(Z_{trn}) > \mathbb{P}(Z_{trn} \,|\, H)\} \\
&= \mathbb{E}_H \big\| \mathbb{P}(Z_{trn}) - \mathbb{P}(Z_{trn} \,|\, H) \big\|_{\mathcal{T}} = 1 - S(Z_{trn}, H) \\
&\geq 1 - \mathbb{S}(\mathcal{L}) - \delta
\end{aligned}
$$

In the last line, we used the fact that $S(Z_{trn}; H) \leq \mathbb{S}(\mathcal{L}) - \delta$ when observations are distributed according to $\mathbb{P}_\delta(z)$. Hence, for any $\delta > 0$, there exists a distribution and a parametric loss function such that:

$$
1 - \mathbb{S}(\mathcal{L}) - \delta \leq |R_{true}(\mathcal{L}) - R_{emp}(\mathcal{L})| \leq 1 - \mathbb{S}(\mathcal{L})
$$

Therefore, algorithmic stability is also necessary for uniform generalization.

## 4 Proof of Theorem 2

Because $\mathcal{Z}$ is countable, we will assume without loss of generality that $\mathcal{Z} = \{1, 2, 3, \ldots, \ldots\}$, and we will write $p_z = \mathbb{P}(Z_{trn} = z)$ to denote the prior (original) distribution of observations. Since all lazy learners are equivalent, we will look into the lazy learner whose hypothesis $H$ is itself the entire training set $S_m$ up to a permutation. Let $m_z$ denote the number of times $z \in \mathcal{Z}$ was observed in the training set. Note that $\mathbb{P}(Z_{trn} = z|H) = \mathbb{P}_{S_m}(z)$, and so $S(H; Z_{trn}) = 1 - \mathbb{E}_{S_m} \|\mathbb{P}(z), \mathbb{P}_{S_m}(z)\|_{\mathcal{T}}$.

We have:

$$
\mathbb{P}(H) = \mathbb{P}(S_m) = \binom{m}{m_1, \, m_2, \, \ldots} p_1^{m_1} \, p_2^{m_2} \cdots
$$

Here, $\binom{\cdot}{\cdot}$ is the multinomial coefficient. Using the relation $\langle \mathbb{P}(X), \, \mathbb{P}(Y) \rangle = 1 - \frac{1}{2} \|\mathbb{P}(X) - \mathbb{P}(Y)\|_1$, we obtain:

$$
\begin{aligned}
S(H; Z_{trn}) &= 1 - \frac{1}{2} \, \mathbb{E}_H \, \|\mathbb{P}(Z_{trn}) - \mathbb{P}(Z_{trn}|H)\|_1 \\
&= 1 - \frac{1}{2} \times \sum_{k=1,2,3,\ldots} \; \sum_{m_1+m_2+\ldots=m} \binom{m}{m_1, m_2, \ldots} \times p_1^{m_1} p_2^{m_2} \cdots \left| \frac{m_k}{m} - p_k \right|
\end{aligned}
$$

For the inner summation, we write:

$$
\begin{aligned}
&\sum_{m_1+m_2+\ldots=m} \binom{m}{m_1, \, m_2, \, \ldots} p_1^{m_1} \, p_2^{m_2} \cdots \left| \frac{m_k}{m} - p_k \right| \\
&= \sum_{s=0}^{m} \binom{m}{s} p_k^s \left| \frac{m_k}{m} - p_k \right| \times \\
&\qquad \sum_{m_1+\ldots+m_{k-1}+m_{k+1}+\ldots=m-s} \binom{m-s}{m_1, \ldots, m_{k-1}, m_{k+1}, \ldots} \times p_1^{m_1} \cdots p_{k-1}^{m_{k-1}} \, p_{k+1}^{m_{k+1}} \cdots
\end{aligned}
$$

Using the multinomial series, we simplify the right-hand side into:

$$
\sum_{s=0}^{m} \binom{m}{s} p_k^s \, (1 - p_k)^{m-s} \left| \frac{s}{m} - p_k \right|
$$

Now, we use *De Moivre's formula* for the mean deviation of the binomial random variable (see the proof of Example 1). This gives us:

$$\sum_{m_1+m_2+\ldots=m} \binom{m}{m_1,\, m_2,\, \ldots} p_1^{m_1}\, p_2^{m_2} \cdots \left| \frac{s}{m} - p_k \right|$$

$$= \sum_{s=0}^{m} \binom{m}{s} p_k^s \, (1-p_k)^{m-s} \left| \frac{s}{m} - p_k \right|$$

$$= \frac{2}{m} \, (1-p_k)^{(1-p_k)m} p_k^{1+mp_k} \frac{m!}{(p_k m)! \, ((1-p_k)m - 1)!}$$

Using *Stirling's approximation* to the factorial [17], we obtain the simple asymptotic expression:

$$\sum_{m_1+m_2+\ldots=m} \binom{m}{m_1,\, m_2,\, \ldots} p_1^{m_1}\, p_2^{m_2} \cdots \left| \frac{m_k}{m} - p_k \right| \sim \sqrt{\frac{2p_k(1-p_k)}{\pi m}}$$

Plugging this into the earlier expression for $S(H; Z_{trn})$ yields:

$$S(H; Z_{trn}) \sim 1 - \frac{1}{2} \sum_{k=1,2,3,\ldots} \sqrt{\frac{2p_k(1-p_k)}{\pi m}}$$

$$= 1 - \sqrt{\frac{\mathbf{Ess}\left[\mathcal{Z};\, \mathbb{P}(z)\right] - 1}{2\pi m}}$$

Due to the tightness of the Stirling approximation, the asymptotic expression for mutual stability is tight. Because $S(H; Z_{trn}) = 1 - \mathbb{E}_{S_m} ||\mathbb{P}(z),\, \mathbb{P}_{S_m}(z)||_{\mathcal{T}}$, we deduce that:

$$\mathbb{E}_{S_m} ||\mathbb{P}(z),\, \mathbb{P}_{S_m}(z)||_{\mathcal{T}} \sim \sqrt{\frac{\mathbf{Ess}\left[\mathcal{Z};\, \mathbb{P}(z)\right] - 1}{2\,\pi\,m}},$$

which provides the asymptotic rate of convergence of an empirical probability mass function to the true distribution.

## 5   Proof of Theorem 4

Let $c_H(z) = \mathbb{I}\{\mathbb{P}(Z_{trn} = z | H) \geq \mathbb{P}(Z_{trn} = z)\}$. We have:

$$\mathbb{S}(\mathcal{L}) = \inf_{\mathbb{P}(z)} S(H; Z_{trn}) = \inf_{\mathbb{P}(z)} \left\{ \mathbb{E}_H \sum_{z \in \mathcal{Z}} \min\{\mathbb{P}(Z_{trn} = z), \mathbb{P}(Z_{trn} = z | H)\} \right\} \tag{5}$$

$$= \inf_{\mathbb{P}(z)} \left\{ 1 - \mathbb{E}_H \sum_{z \in \mathcal{Z}} \left( \mathbb{P}(Z_{trn} = z | H) - \mathbb{P}(Z_{trn} = z) \right) \cdot c_H(z) \right\} \tag{6}$$

$$= 1 - \sup_{\mathbb{P}(z)} \left\{ \mathbb{E}_H \sum_{z \in \mathcal{Z}} \left( \mathbb{P}(Z_{trn} = z | H) - \mathbb{P}(Z_{trn} = z) \right) \cdot c_H(z) \right\} \tag{7}$$

$$= 1 - \sup_{\mathbb{P}(z)} \left\{ \mathbb{E}_{S_m} \mathbb{E}_{H|S_m} \left[ \mathbb{E}_{Z \sim \mathbb{P}(z)}\, c_H(Z) \; - \; \mathbb{E}_{Z \sim S_m}\, c_H(Z) \right] \right\} \tag{8}$$

$$\geq 1 - \sup_{\mathbb{P}(z)} \left\{ \mathbb{E}_{S_m} \mathbb{E}_{H|S_m} \left| \mathbb{E}_{Z \sim \mathbb{P}(z)}\, c_H(Z) \; - \; \mathbb{E}_{Z \sim S_m}\, c_H(Z) \right| \right\} \tag{9}$$

$$\geq 1 - \sup_{\mathbb{P}(z)} \left\{ \mathbb{E}_{S_m} \sup_{h \in \mathcal{H}} \left| \mathbb{E}_{Z \sim \mathbb{P}(z)}\, c_h(Z) \; - \; \mathbb{E}_{Z \sim S_m}\, c_h(Z) \right| \right\} \tag{10}$$

Next, we note that the quantity inside the expectation in (10) can be bounded using uniform convergence. In particular, we use the following bound, which holds for any distribution $\mathbb{P}(z)$ if $m > d_{VC}(\mathcal{C}) + 1$[1]:

$$\mathbb{E}_{S_m} \sup_{h \in \mathcal{H}} \left| \mathbb{E}_{Z \sim \mathbb{P}(z)}\, c_h(Z) \; - \; \mathbb{E}_{Z \sim S_m}\, c_h(Z) \right| \leq \frac{4 + \sqrt{d_{VC}(\mathcal{C})\,(1 + \log(2m))}}{\sqrt{2m}}$$

## Footnotes

[1] A proof of this bound is in Eq. 6.4 and Lemma 6.10 in the textbook *"Understanding Machine Learning: From Theory to Algorithms"* by Shai Shalev-Shwartz and Shai Ben-David, 2014.