[Reviews · NeurIPS 2015]

Submitted by Assigned_Reviewer_1

Note: References present in the paper are referred to by their numerical citations as used in the paper.

Summary of Paper ======================================== The paper seeks to establish a connection between algorithmic stability and generalization performance. Notions of algorithmic stability have been proposed before and linked to the generalization performance of learning algorithms [6,11,13,14] and have also been shown to be crucial for learnability [14].

[12] proved that for bounded loss functions, the generalization of ERM is equivalent to the probabilistic leave-one-out stability of the learning algorithm. [14] then showed that a problem is learnable in Vapnik's general setting of learning iff there exists an asymptotically stability ERM procedure.

This paper first establishes that for Vapnik's general setting of learning, a probabilistic notion of stability, is necessary and sufficient for the training losses to converge to test losses uniformly for all distributions. The paper then presents some discussions on how this notion of stability can be interpreted to give results in terms of the capacity of the function class or the size of the population.

Questions ======================================== - The utility of the main result of the paper is not clear - does the paper want to establish the convergence of training to test errors in cases when uniform convergence does not hold? - Can the authors demonstrate one example of a learning problem where uniform generalization is impossible but the training errors of a stable algorithm still converge to the test errors? - Can the results of [14] not be used to establish a similar result? [14] establish that learnability is equivalent to the existence of a stable AERM. Let f* be a population risk minimizer, f_ERM be an empirical risk minimizer, f_AERM be a stable asymptotic empirical risk minimizer as defined in [14]. Let R and R^ denote the population and empirical risk functionals. Then we have

R(f_AERM) <= R(f*) + epsilon (with enough samples since the problem is learnable)

<= R^(f*) + 2*epsilon (point convergence on f* using Hoeffding inequality)

<= R^(f_ERM) + 2*epsilon (f_ERM achieves smallest empirical risk)

<= R^(f_AERM) + 3*epsilon (with enough samples since algorithm is an AERM)

This establishes R(f_AERM) - R^(f_AERM) <= epsilon for any epsilon, for large enough number of samples. This recovers the result this paper is trying to show. - In general, why should one care about the difference of training and test error? It is learnability and test error that one is interested in and the result of [14] already establishes that learnability is equivalent to the existence of uniformly RO-stable asymptotically empirical risk minimizers.

- A thorough discussion detailing the contributions of the paper w.r.t. previous results is thus required. - The paper does not link its notion of stability to existing notions of algorithmic stability - this is necessary since existing notions like uniform RO-stability [14] have already been used to establish similar results. - The discussion in Section 5.1 is vague and results are not stated formally. Dropout is a widely used heuristic and recent theoretical analyses of the method have generated some interest. It seems like a lost opportunity to have a discussion on the theoretical merits of dropout but not state anything formally. - The results in Section 5.2 are not properly instantiated. The notion of ESS is introduced but never used to actually bound effective size of some observation space except for the very simple case of finite spaces in Corollary 1. - The same holds true for Section 5.3 as well. The results are not properly instantiated save the simple case of finite hypothesis classes in Theorem 3. The second result (Theorem 4) establishes a different way of calculating the VC dimension of a function class that is not necessary a classification class. However, yet again, no instantiations are given to demonstrate if this notion of VC dimension is indeed useful and can be linked, for example, to other well known notions of capacity such as uniform entropy, Rademacher averages etc. - As such it becomes difficult to assess if the results in Section 5 are of any interest and if they shed any new light on the topic.

Quality ======================================== The paper may have interesting contributions but they are neither discussed, nor properly put in context of existing results. The goal of bounding the difference of training and test errors is not well-motivated, especially when uniform convergence is not possible.

Clarity ======================================== The paper is written well.

Originality ======================================== The paper mostly uses well established techniques in information complexity such as the data processing inequality. The notion of stability via the total variation distance between distributions seems novel and interesting.

Significance ======================================== I am not sure if the paper, in its current form, would terribly excite the learning theory community since algorithmic stability is already know to be equivalent to learnability. The other results in Section 5 are poorly instantiated and it is not clear if they present novel insights into the problem.
Summary: The main contribution of the paper is to establish a connection between algorithmic stability and generalization performance. A new probabilistic notion of algorithmic stability is presented and connections are drawn to the generalization performance of learning algorithms. The paper is self-contained but does not link itself very well to the existing results in this area.

Submitted by Assigned_Reviewer_2

SUMMARY

This paper defines a notion of algorithmic stability that is shown to be equivalent to uniform generalization. Different consequences for generalization are explored in connection to techniques used to control algorithmic stability directly, richness of the hypothesis space and techniques used to control it, and also a notion of size of the input space.

CONTRIBUTIONS

* A notion of algorithmic stability that corresponds to uniform generalization. * Theoretical justifications for different popular learning strategies used to improve algorithmic stability. * Strengthening of the understanding of generalization and: the input space, the hypothesis space, the learning algorithm.

These contributions strengthen the understanding of generalization, and can lead to interesting future work that deepens the understanding of different techniques used to increase generalization (hold-out set validation, regularization, dimensionality reduction, etc.). Proposing a notion of stability that is equivalent to (uniform) generalization is challenging, and I think this makes the contributions of this paper strong.

SUPPORT

The proofs are correct, simple and easy to understand. The claims that provide a justification for techniques that improve generalization are well-founded in the theoretical results. The proof sketches are informative as well, while maintaining the pace of the exposition.

TECHNICAL QUALITY

The paper is technically sound, and the consequences of the central results we well-explored. The authors have defined stability on expectation, not with high-probability (in contrast to sample complexity bounds, which hold with high-probability). The expectation allows us to obtain low-probability bounds with Markov's inequality, but it is not clear whether high-probability bounds can be obtained without fixing a specific algorithm.

ORIGINALITY

The paper seems well-inserted in the literature, but my knowledge of related work in the area of stability is shallow and I cannot make a credible assessment of originality.

CLARITY

The paper is well-written, with clear explanations and clear proofs. This theoretically-oriented work is well executed and it is an accessible read (in this reviwer's opinion). Based on the appreciation of other reviewers, though, it seems that the clarity of the paper can be improved.

FOR THE REBUTTAL

Please comment on the issue of expectation bounds vs. high-probability bounds. Can high-probability bounds be obtained without fixing an algorithm, or are the expectation bounds alone a significant contribution?

It would be good if the authors could comment on the relationship between their definition of stability and existing definitions, this has been raised by other reviewers too.

DETAILED COMMENTS

The definition of stability proposed is interesting; it is sensible, as it can be interpreted as "the output of the algorithm does not depend too much on any single example", but I was wondering how it relates to existing definitions (e.g., the ones where one observation in the sample is dropped, or where it is swapped for another one). On the other hand perhaps the point is not so much to compare them, in case the endeavour in previous works has been to characterize generalization with some notion of stability.

Considering Section 5.3, do you think we should be caring about Shannon Entropy of the output hypothesis (when designing algorithms), rather than the size of the hypothesis space? I am asking this because of the issues with bad ERM learners in multiclass classification [1], in which case we have to do something besides "just ERM" to be successful.

Example 2 made me think of hold-out set validation (where a validation set is used to increase the "stability" of the algorithm) and also of minimization using Bregman projections (minimize a convex loss, then do a Bregman projection on a set) -- it also seems similar to post-processing.

I see that the concepts in Definition 8 match the losses you use to show that stability is necessary for uniform generalization. Do you think it allows you to conclude that classification (with the 0-1 loss) is a hardest problem from the point of view of stability/generalization?

REFERENCES:

[1] Daniely, Sabato, Ben-David & Shalev-Shwartz (2013). "Multiclass learnability and the ERM principle."

FIXES

[l:60] "nature of the observation" [l:127] I am not a fan of this notation, why not use conditional expectation? (Using the subscript of the expectation only to indicate the distribution of some r.v. is helpful, though.) [l:152] The symbolic convention somewhat breaks when you use $\mathcal{L}$ for the algorithm, so why not use simply $A$? [l:173] I would suggest removing the last sentence from this paragraph. [l:190-191] "variables are independent of each" [l:265-266] "H_2 contains less information" [l:317] Here and in the appendix you used $\mathcal{V}$ to denote the total variation, while you defined it with a $\mathcal{T}$. [l:357-358] "always holds:" [l:383] You inverted the arguments of the KL divergence here. [l:400] $\mathbb{P}(Z_{\mathrm{trn}} = z)$ (the parentheses need to be fixed). [l:420] I would suggest using verbs in the past here "In this paper, we showed" [l:421-422] "always vanishes at"

(Appendix) [l:185] (double ellipsis) [l:228] Perhaps you can add the reference to H. Robbins (1955) and Shalev-Swartz & Ben-David's book at the end of the appendix, or expand the one to H. Robbins (1955) as a footnote.

POST-REBUTTAL REMARKS

There seems to be little understanding of the connections between stability by conditioning vs. stability by observation elimination vs. stability by observation switching. It would seem that the definition arose as a tool that worked for the desired results, and the authors should be upfront about the limited understanding of the aforementioned connections.

The authors suggest that a strength of their main result is its algorithm-independence. However this algorithm-independence is the case only for expectation guarantees on stability, not high-probability guarantees, which are often what one is interested in. On the other hand, it is known how to convert expectation guarantees for a base algorithm into high-probability guarantees for the same algorithm combined with a hold-out set validation scheme (cf. Shalev-Shwartz & Ben-David, "Understanding Machine Learning: From Theory to Algorithms"). Therefore, the authors should mention this limitation of their results. The contributions are nevertheless solid and interesting -- of course, not as strong as algorithm-independent high-probability guarantees, which could be pursued as future work.
Summary: This is an exciting paper, the text is well-written, and interesting results are proved with simple techniques, which lends elegance to the results. However, there seem to be some unaddressed limitations of the results.

Submitted by Assigned_Reviewer_3

SUMMARY: Following the traditional approach to algorithmic stability pioneered in [6], the authors study the relationship between generalization and stability. However, contrary to previous approaches, they consider the general setting of learning. The result is interesting because they have necessary and sufficient condition to (uniform) generalization, not just bounds. However, the main drawback is the "uniform" part: it is not clear if it is a too strong condition, i.e. being able to generalize wrt to *any* parametric bounded loss.

DETAILED COMMENTS: The paper is very interesting and clearly written.

In a sense, as the authors' say, this work complements the results in [14], finding necessary and sufficient conditions to generalization, rather than learnability, in the general setting of learning.

I have to say that there has been always some kind of division in the ML community about people that care about learnability and the ones that care about generalization. Personally, I find both interesting and different enough to generate different approaches and papers. Hence, the topic is the paper is clearly interesting, at least for a part of the NIPS community, even if it does not cope with learnability.

There are also some drawbacks in this paper. The main problem for me is the lack of depth in the discussion of the implications (see for example Remarks 1-2 and Examples 2-3). The only clear case seems to be the one for finite VC dimension and you also get a feeling of mild overselling in reading some remarks and comments. This lack of depth and the unconventional (for a part of the ML community) definition of the algorithmic stability could make the paper accessible to a small niche of people and reduce its impact. This is also important because the definition of uniform generalization (def. 3) appears quite strong, and it can be only justified if we buy the authors' thesis that the equivalent algorithmic stability (def. 5) is "rather weak". That is to say, most of the algorithms will satisfy it, so we have that they will generalize, even wrt any bounded loss. However, this "weakness" is stated only in vague and intuitive terms, rather than in a more ML-friendly way.

This seems problematic, but I don't think this can be a reason for rejection.

In fact, as I understand it, the loss function here has no relationship with the algorithm used. Indeed, the algorithm itself does not have to be ERM nor use it in any part of it. Overall, it seems to me that there are no concrete elements to judge the condition of "uniform generalization" as "too strong", besides one's intuition. On the other hand, the condition of algorithmic stability does appear weak to me, and, more importantly than any "feeling" and/or "intuition", seems to have nice representation in some cases (e.g. finite VC dimension).

Another problem, as also found by the other reviewers, is the fact that the bounds are only in expectation, but I find this a minor thing. Indeed, many of the results based on algorithmic stability are only in expectation, yet they have succeeded in inspiring new algorithms and techniques.

If the paper is accepted, I encourage the authors to provide more detailed examples and/or quantification of the algorithmic stability in other interesting cases, if only to increase the impact of their work.

Overall, I find the result correct, novel and mature enough to be published in NIPS.
Summary: The paper presents a necessary and sufficient condition to (uniform) generalization, through a novel notion of algorithmic stability in the generic setting of learning. A few implications and examples are shown as well.

I found the paper clearly written and interesting enough to be accepted, even if a more in-depth discussion of the implications is needed.

Submitted by Assigned_Reviewer_4

Abstract

The authors define uniform generalization in the sense that |\hat R_emp(L(h_D)) - \hat R(L(h_D))| -> 0

for all distributions P^n generating the data D in an i.i.d fashion and all loss functions L, where: - h_D is the hypothesis of the algorithm for the data D - \hat R_emp is the expected (w.r.t D~P^n) empirical risk

with respect to the loss L, and \hat R is the expected

true risk

(I hope I got this right, since their notation is not really

clear at this point). The definition of algorithmic stability is even more cryptic, please see Definition 5. In any case, the main Theorem 1 shows that both notions are

equivalent. Theorem 3 then shows that finite VC dimension implies their notion of algorithmic stability and some

addiitonal results are provided.

Comments

As my attempt to summarize the paper already indicates

I had a somewhat hard time reading the paper, in particular since somewhat strange notations are used. This complicated appearance is in contrast to the rather simple ideas, see e.g. the proof of Theorem 1. In any case, my major concerns regarding the paper is that

the notions of genralization and stability are not really the ones we are usually interested in. For example, let us look at uniform generalization: If I want to understand an algorithm for least squares regression, I usually do not care

whether this algorithm also works for, say classification.

But this is exactly assumed in uniform generalization. In addition, I do not understand, why expected risks are

considered. Generalization is usually considered in a

"with high probability" setting and expectations are usually only taken to significantly simply considerations.

In the case of algorithmic stability, which should better be called uniform algorithmic stability since an infimum is taken over all distributions, I could not really get an intuition

about its meaning at all, and the few examples, the authors

provide are not helpful in this regard either (exception: Theorem 3, which is helpful).

Minor comments: - The way expectations are denoted makes it hard to understand

what is meant. There are better ways to denote such things

without extra effor. - The "Markov chain notation" is also rather confusing. Again,

there is no need to use this kind of notation.
Summary: The paper compares a certain notion of algorithmic stability with

a certain form of uniform generalization. The main result shows that both notions are equivalent. In addition, a few

rather concrete cases are worked out in detail.

Submitted by Assigned_Reviewer_5

(This is a "heavy" review.)

== SUMMARY ==

This paper proves that a certain notion of algorithmic stability (in the learning algorithm) is a necessary condition for "uniform generalization". In the paper, "algorithmic stability" is a probabilistic definition, which loosely means that the (random) hypothesis output by a (randomized) learning algorithm should have decreasing dependence on any single example as the size of the training set approaches infinity. In this context, uniform generalization means that there exists a minimum training size such that the difference of the empirical and expected losses (in expectation over hypotheses output by the learning algorithm) is upper-bounded, uniformly over all "parametric" loss functions and data distributions. Uniform generalization is a stronger condition than regular generalization -- which, using the paper's definition, only needs to hold asymptotically, for a given loss function. The main theorem is interpreted in various ways: to explain the benefit of methods like dimensionality reduction and dropout; to analyze the relationship between the effective size of the domain and algorithmic stability (hence, generalization); and to make connections between algorithmic stability and VC dimension.

All in all, I find this to be a well-crafted, insightful theory paper. There are well-placed examples that ground the theory in practice, and the implications of the main theorem are interesting. (I especially liked the analysis of the effective size of the domain.) The writing could use some minor polishing (see detailed notes), but, overall, it's pretty good.

My only concern is that it probably won't have much impact for practitioners; there are no take-away messages, no "prescriptions". The paper gives us a better understanding of learning (in theory), but it doesn't propose anything that we should be doing differently (in practice). It would be a much stronger paper if its insights motivated a new technique.

== HIGH-LEVEL COMMENTS ==

I was thrown off by the term "inference process" used in the abstract. This usually refers to prediction (i.e., infer Y given X), but it is equivalent to learning in this paper (i.e., infer H given S_m). The paper could be clearer about this distinction, or (my preference) you could just avoid calling it inference. On a related note, it would interesting to analyze whether stability in the predictor's inference is a necessary condition for generalization.

The main result hinges on the subtle distinctions between "learnability", "consistency", "uniform convergence" and "(uniform) generalization". I found the discussion of these concepts (Sec 2) to be a bit tortured and would have appreciated some discussion of why these distinctions matter. Is there a class of learning algorithms (or hypotheses) that only supports learnability but not generalization (or vice versa)? Why should the reader care about these distinctions?

On a related note, the characterization of "learnability" via the excess risk seems strange to me. I've never read this definition anywhere. Normally (to me), learnability means PAC learnability; that is, there exists a poly(\epsilon,\delta,m)-time algorithm such that, with probability at least 1-\delta over draws of m examples, the learned hypothesis has error at most \epsilon. This definition doesn't appear to be equivalent to yours. I guess what I'm saying is you should make it clearer that the definition you're using is taken from source XYZ, and may not be the definition that the reader is accustomed to.

I was a bit confused by the application of the data processing inequality in the proof of Theorem 1. If the Markov chain is

( S_m -> H -> L(.;H) ) == ( A -> B -> C ) then what is Z_trn? Is it considered A in this construction? The inequalities on line 242 seem to use the fact that

S(L(.;H);Z_trn) > S(H;Z_trn), but doesn't this mean that S(C;A) > S(B;A), and isn't this a different inequality from Lemma 1?

== DETAILED COMMENTS ==

- Lines 19 and 50 : "new novel" -> "new". - Line 25 : "justification to" -> "justification for". - Lines 34 - 44 read in like a tutorial on machine learning, which is a bit too rudimentary for the intended audience. - Line 40 : "such two objectives" -> "these two objectives". - Line 50 : "example to such approach is the development of the SVM" -> "example of such an approach is the SVM". - Line 52 : "subtle" is the wrong word here, but I can't figure out what is meant here, so I can't suggest a better word. - Line 55 : "such rich theories" -> "these rich theories". - Line 80 : "such result" -> "this result". - Line 107 : I've only ever seen generalization defined for a given hypothesis class; never for a learning algorithm. Can you provide a citation for this definition? - Line 197 : "we define stability" -> "we define the stability". - Line 302 : "bleak" seems like too strong a word. - Line 312 : "such notion" -> "this notion". - Line 422 : "vanish" -> "vanishes". - Line 424 : "choice of the" -> "choice of". - Line 425 : "such result" -> "this result".

== POST-RESPONSE ASSESSMENT ==

I'm still not convinced that this paper will have any impact. The relationship between stability and generalization has been well-known for over a decade, so the high-level message is nothing new. This paper basically just expands stability theory and confirms a conjecture for a more general learning setting. The results don't motivate any specific new technique, and the implications for existing data processing heuristics (Ex. 2 & 3) are a bit hand-wavy. Nonetheless, it's a well-written paper, and an interesting read from the theory perspective; just because I don't see the practical value does not mean that others won't. Therefore, I recommend acceptance.
Summary: The paper proves that algorithmic stability is a necessary condition for uniform generalization. The theory seems technically sound (although a bit confusing), the insights are intriguing, and the presentation is good, but the result probably won't have much impact.

Author Feedback
Author rebuttal: We thank the reviewers for their valuable feedback, and we apologize for not being able to respond to every comment due to the space limit. Kindly note that all typing/grammatical errors will be fixed.

Reviewer_1:
-"Can the results of [14] not be used to establish a similar result?"
The results of [14] show that learnability is equivalent to the existence of a stable AERM. In our case, we consider the question of when generalization holds for an arbitrary algorithm, which may or may not be an AERM. As alluded to in Section 2, we believe our result complements that of [14]. In [14], it is shown that stability is key to learnability via the notion of AERM. In our case, we show that stability is key to generalization.

-"The discussion in Section 5.1 is vague"
These results follow by direct application of the data processing inequality, which is proved formally in Lemma 1. It explains quite intuitively why dropout and other techniques can improve generalization although we have not quantified the gain yet. We agree that quantifying the gain of such methods indeed deserves attention in the future. In this paper, however, we only wanted to bring to attention the utility of the proposed notions of stability and uniform generalization in analyzing such techniques.

Reviewer_2:
-"It would be good if the authors could comment on the relationship between their definition of stability and existing definitions"
Our definition of stability is information-theoretic, which does not resemble any previous work in the literature. It allows us to link stability to many factors such as the domain, the hypothesis space & VC dimension, and mutual information. We agree that it would be interesting to provide further links to existing notions of stability, such as those based on the leave-one-out method, and this is the subject of an ongoing work.

-"Considering Section 5.3, do you think we should be caring about Shannon Entropy of the output hypothesis rather than the size of the hypothesis space?"
We believe this is indeed the case since entropy is shown to give a tighter (more accurate) generalization bound than the size of the hypothesis space.

Reviewer_3:
-"My only concern is that it probably won't have much impact for practitioners"
Perhaps, the main take-away message when designing learning algorithms is how stability is key to generalization. Also, a second take-away message is how data processing mitigates overfitting. Currently, we're investigating how this can be employed to improve extreme multiclass classification via post-processiong at the output layer. This is an ongoing work.

-"the characterization of 'learnability' via the excess risk seems strange"
This definition is available, for instance, in [14,17]. It is usually referred to as "agnostic PAC learnability", which reduces to PAC learnability in the realizable setting.

-"doesn't this mean that S(C;A) > S(B;A), and isn't this a different inequality from Lemma 1?"
If S_m - > H - > L(.,H) holds, then Z_trn - > H - > L(.,H) holds as well because Z_trn is drawn from S_m uniformly at random. Hence, using the notation of Lemma 1, A=Z_trn. The inequality S(C;A) > S(B;A) is precisely the statement of Lemma 1. On a related note, because A - > B - > C is equivalent to the reverse chain C - > B - > A, we also have S(B;C) < S(A;C), which is used in Section 5.1 (line 282).

Reviewer_6:
-"I had a somewhat hard time reading the paper, in particular since somewhat strange notations are used"
This is quite unfortunate! We've striven to make the paper as accessible and self-contained as possible. The notation is fully explained in Section 3. They are also used in the standard textbook "Elements of Information Theory" by T. Cover and J. Thomas [19].

-"If I want to understand an algorithm for least squares regression, I usually do not care whether this algorithm also works for, say classification. But this is exactly assumed in uniform generalization"
True. However, our main results do not dictate that one has to 'plan' for uniform generalization when designing an algorithm. Rather, it states that it comes as a byproduct of (in fact equivalent to) algorithmic stability. Of course, this implies that stability, in the weak sense given in Definition 5, is always a sufficient condition for generalization in the traditional sense. In addition, there are cases where uniform generalization might be useful, such as in prototype learning and density estimation whose learned hypothesis can be used for classification, regression, and clustering (i.e. with multiple parametric loss functions). If the learning algorithm is stable, then uniform generalization justifies using the same hypothesis in many applications because the generalization risk is guaranteed to vanish "uniformly" across all parametric loss functions.